Quantitative real-time PCR analysis of bacterial biomarkers enable fast and accurate monitoring in inflammatory bowel disease

Sezgin Efe efesezgin@iyte.edu.tr 1
Terlemez Gamze 1
Bozkurt Berkay 1
Bengi Göksel 2
Akpinar Hale 2
Büyüktorun İlker 2
1 Izmir Institute of Technology , Izmir , Turkey
2 Dokuz Eylül University , Izmir , Turkey
Saki Morteza
Electronic publication date: 2022 Oct 18
Publication date: 2022
Volume: 10
Electronic Location ID: e14217
Received 2022 Jun 27; Accepted 2022 Sep 20
Copyright: ©2022 Sezgin et al.
Copyright year: 2022
Copyright holder: Sezgin et al.
License: This is an open access article distributed under the terms of the Creative Commons Attribution License, which permits unrestricted use, distribution, reproduction and adaptation in any medium and for any purpose provided that it is properly attributed. For attribution, the original author(s), title, publication source (PeerJ) and either DOI or URL of the article must be cited.
License URL: https://creativecommons.org/licenses/by/4.0/

Keywords: Quantitative real-time PCR, Molecular biomarker, Inflammatory bowel disease, Crohn’s disease, Ulcerative colitis

Funding: Turkish Society of Gastroenterology IZTECH Scientific Research Projects Committee (BAP) IYTE0209 This work was supported by the Turkish Society of Gastroenterology, and the IZTECH Scientific Research Projects Committee (BAP) (project number IYTE0209). There was no additional external funding received for this study. The funders had no role in study design, data collection and analysis, decision to publish, or preparation of the manuscript.

==============================
Inflammatory bowel diseases (IBD) affect millions of people worldwide with increasing incidence. Ulcerative colitis (UC) and Crohn’s disease (CD) are the two most common IBDs. There is no definite cure for IBD, and response to treatment greatly vary among patients. Therefore, there is urgent need for biomarkers to monitor therapy efficacy, and disease prognosis. We aimed to test whether qPCR analysis of common candidate bacteria identified from a patient’s individual fecal microbiome can be used as a fast and reliable personalized microbial biomarker for efficient monitoring of disease course in IBD. Next generation sequencing (NGS) of 16S rRNA gene region identified species level microbiota profiles for a subset of UC, CD, and control samples. Common high abundance bacterial species observed in all three groups, and reported to be associated with IBD are chosen as candidate marker species. These species, and total bacteria amount are quantified in all samples with qPCR. Relative abundance of anti-inflammatory, beneficial Faecalibacterium prausnitzii, Akkermansia muciniphila, and Streptococcus thermophilus was significantly lower in IBD compared to control samples. Moreover, the relative abundance of the examined common species was correlated with the severity of IBD disease. The variance in qPCR data was much lower compared to NGS data, and showed much higher statistical power for clinical utility. The qPCR analysis of target common bacterial species can be a powerful, cost and time efficient approach for monitoring disease status and identify better personalized treatment options for IBD patients.

Introduction

Inflammatory bowel diseases (IBD) are complex, heterogeneous diseases arising from chronic and uncontrolled inflammation of the gastrointestinal (GI) tract (Podolsky, 2002; Tontini et al., 2015). Microbiota, genetics, and environmental factors are suggested to be underlying factors for susceptibility to IBD (Albenberg, Lewis & Wu, 2012). Ulcerative colitis (UC) and Crohn’s disease (CD) are the two most common diseases categorized under IBD. Accurate IBD diagnosis requires examination of clinical, endoscopic, and histopathological characteristics, but none of the findings are definitive, and even some patients’ differential diagnoses cannot be made. IBD has become a worldwide disease affecting millions of patients (Alatab et al., 2020). The biggest incidence ratios have been reported in Northern Europe and North America for CD and UC (Burisch & Munkholm, 2015).

IBD has important social, psychological and financial implications as well as the deterioration of health-related quality of life. IBD impresses personal life and imposes significant economic burden not only on the patient but also on the health care system, due to treatment costs, time lost from work, and reduced productivity at work (Mehta, 2016; Walter et al., 2020). The financial burden can be even higher as IBD also affects individuals at an early age. Given the big personal and cumulatively population level costs, there is great interest in identifying both useful biomarkers and techniques to assay these markers for IBD progression, therapy response, and control.

The definite cause of IBD is not known, so individual or population level biomarker screens to identify people at risk are not possible yet. Most IBD patients seek medical care at later, more advanced stages of the disease, and early intervention to prevent disease progression is rare. Therefore, identification of biomarkers, and techniques to assay these markers to monitor therapy efficacy, and disease prognosis is of great importance.

Although a number of biomarkers are suggested for the diagnosis of IBD, however with questionable sensitivity and specificity for UC and CD (Soubieres & Poullis, 2016; Chen et al., 2020; Guo et al., 2021), biomarkers for monitoring disease progression or response to therapy is lacking and development of such biomarkers is an active research area. Serological (Miranda-Garcia, Chaparro & Gisbert, 2016), metabolomic (Bjerrum et al., 2017; Keshteli et al., 2018; Notararigo et al., 2021), proteomic (Kalla et al., 2021), metagenomic (Zhou et al., 2018; Serrano-Gomez et al., 2021), and transcriptomic (Montero-Melendez et al., 2013) approaches have been reported. However, these large data driven ‘omics’ techniques are research based, rather expensive, time consuming, and their clinical utility is questionable. Therefore, faster, cheaper, more accurate techniques that can utilize already available equipment in hospitals or molecular diagnostic laboratories is necessary. Discovery of bacterial biomarkers by next generation sequencing and further quantification of selected bacterial species by quantitative real time PCR (qPCR) is well documented in literature (Machiels et al., 2014; Lopez-Siles et al., 2020; Mondot et al., 2016; Zhou et al., 2016; Lopez-Siles et al., 2014; Lopez-Siles et al., 2018; Pascal et al., 2017). Therefore, qPCR can be a highly efficient molecular tool for monitoring IBD progression and response to therapy.

Based on the urgent need for such reliable methods with possible clinical utility, we aimed to test whether qPCR analysis of common candidate bacterial species identified from a patient’s individual fecal microbiome can be used as a fast and reliable personalized microbial biomarker for efficient monitoring of IBD. We focused on Turkish IBD patients because Turkey is among the countries with highest increase of IBD incidence (Can et al., 2019), but microbiota studies from Turkey are rather limited, and the utility of fast and accurate molecular techniques in IBD monitoring has not been explored in this population. In addition to testing previously reported bacterial markers, we searched for novel bacterial species, and evaluated S. thermophilus as a biomarker in IBD.

Material and Methods

Sample collection

Fecal and blood samples were collected from 18 IBD patients (six diagnosed with ulcerative colitis and 12 diagnosed with Crohn’s disease) and four healthy (control group) individuals in the Gastroenterology Department of Dokuz Eylul University Hospital (Izmir, Turkey). IBD patients (ulcerative colitis and Crohn’s disease) were diagnosed according to international guidelines based on clinical, endoscopic, histopathological, and radiological examinations (Stange et al., 2008; Bemelman et al., 2018). The healthy controls were candidates without any history of IBD or mucosal lesions in colonoscopy. Fecal samples gathered in sterile and airtight containers, and blood samples collected into EDTA tubes were transported to laboratory within six hours after collection. All study samples were kept at −80 °C until processing. Ethical approval was obtained from the Ethics Committee of the Dokuz Eylül University (2017/08-03). All participants provided informed consent in the format required by the Dokuz Eylül University ethics committee.

Genomic DNA extraction from blood samples

DNA was isolated from blood samples with the Genomic DNA Mini Kit (blood/cultured cell) (Geneaid Biotech Ltd., Taiwan) following the manufacturer’s protocol. The quality of extracted DNAs (A260/A280 and A260/A230 ratios) was checked with a Nanodrop 8000c Spectrophotometer (Thermo Fisher Scientific, Waltham, MA, USA).

Primer design for candidate variants in IBD associated candidate genes

We targeted rs2066844 (Arg702Trp), rs2066845 (Gly908Arg), rs2066847 (Leu1007insC) SNPs for the NOD2 gene; rs11209026 (Arg381Gln) for the IL-23R gene, and rs2241880 (Thr300Ala) for the ATG16L1 gene. DNA sequences for each gene were obtained from NCBI (https://www.ncbi.nlm.nih.gov/), ENSEMBL (https://www.ensembl.org/index.html), and UCSC Genome Browser (https://genome.ucsc.edu/cgi-bin/hgGateway) databases. These gene sequences were used to design polymerase chain reaction (PCR) primers using the IDT SciTools PCR algorithm (Integrated DNA Technologies) (Table S1). Oligonucleotide properties, melting temperature, hairpins, dimers, and mismatches were identified by IDT SciTools OligoAnalyzer 3.1 (https://www.idtdna.com/calc/analyzer) software (Owczarzy et al., 2008), and specificity of primers were confirmed with Primer-BLAST (https://www.ncbi.nlm.nih.gov/tools/primer-blast/).

Amplification and sequencing of IL-23R, ATG16L1 and NOD2 variants

With primers designed for the specified SNPs, PCR analysis was performed using the FastStart High Fidelity PCR System, dNTPack kit (Roche Applied Science, Penzburg, Germany). Reaction mixes were made separately for IL-23R, NOD2, and ATG16L1 genes in a final volume 25 µl of using 17.25 µl PCR-grade water, 0.5 µl forward and reverse primers, 0.5 µl PCR Grade Nucleotide Mix, 2.5 µl FastStart High Fidelity Reaction Buffer, 0.5 µl dimethyl sulfoxide (DMSO), 0.25 µl FastStart High Fidelity Enzyme Blend, and 3 µl DNA. The thermal cycling was subjected to the following conditions: denaturation at 94 °C for 10 min followed by 35 cycles of 94 °C for 2 min, annealing at 57 °C for 30 s, and elongation at 72 °C for 1 min using SimpliAmp Thermal Cycler (ThermoFisher Scientific, Waltham, MA, USA).

PCR products were verified by agarose gel electrophoresis. Briefly, 5 µL of PCR product was mixed 1 µL 6X DNA loading buffer, and run on 1.4% agarose gel in 0.5X TBE buffer under a steady voltage of 100 V for 60 min at room temperature.

PCR products were purified using ExoSAP-IT™ PCR Product Cleanup Reagent (Thermo Fisher Scientific, Waltham, MA, USA). 10 µl sequencing mixture contained 4 µl ddH2O, 1 µl 5X ABI Buffer, 1 µl of primer (3.2 pmol/µl), 2 µl BigDye Ready Reaction Mix, and 2 µl of the PCR product. Samples were sequenced on ABI Prism 3130xl Genetic Analyzer (Applied Biosystems, Waltham, MA, USA). Sequencing results (ABI chromatograms) were analyzed in Unipro UGENE v.33 (Okonechnikov et al., 2012) program. Multiple sequence alignments using ClustalW algorithm were performed in Unipro UGENE v.33 program (Okonechnikov et al., 2012).

Bacterial DNA isolation from stool samples

DNA was extracted from stool samples using the QIAamp DNA Stool Mini Kit (Qiagen, Hilden, Germany) according to the manufacturer’s protocol. The concentration (ng/µL) and purity (A260/A280 and A260/A230 ratios) of the DNA samples were determined by Nanodrop 8000c Spectrophotometer (Thermo Fisher Scientific, Waltham, MA, USA).

16S rRNA gene amplicon sequencing by next generation sequencing (NGS)

Nine stool samples (three samples from each UC, CD patients, and healthy volunteers) were chosen for amplicon analysis. The variable V3-V4 region of 16S rRNA gene was targeted and amplified with the following PCR primers: 341F (5′-CCTAYGGGRBGCASCAG-3′) and 806R (5′-GGACTACNNGGGTATCTAAT-3′). After the purification of PCR products, sequencing libraries were generated with Nextera XT DNA Library Preparation Kit (Illumina, San Diego, CA, USA). The concentration of sequencing libraries are standardized to 4nM each. Normalized samples were pooled and sequenced by Illumina NovaSeq 6000 as paired-end (2 × 250 bp) using the manufacturer’s standard procedure. Raw data quality control check was performed by FastQC, and quality control of the reads was checked by QIIME2 (Caporaso et al., 2010). Effective tags were obtained after removing primer and barcode sequences, chimeric reads, and reads with Phred Score less than 20 by DADA2 (Callahan et al., 2016). By utilizing the effective tags, representative sequence for each Operational Taxonomic Units (OTUs) were acquired with ≥97% similarity against the Greengenes and SILVA databases. QIIME2 was used for taxonomic determination of each OTU. Rarefaction curves plotting sequencing depth vs. number of taxa identified were used to judge the appropriateness of sequencing depth for each sample (Pereira-Marques et al., 2019; Zaheer et al., 2018).

Species diversity within samples were assessed by five different Alpha diversity estimates including observed-species, Chao1, Shannon, Simpson, and ACE indices. Alpha diversity indices were visualized via boxplots. All downstream analyzes were performed with “phyloseq” (McMurdie & Holmes, 2013) and “ggplot2” packages in R software (Version 4.0.5) (https://www.r-project.org).

Quantification of selected bacterial species levels using Real-Time quantitative PCR (qPCR) analysis

Six bacterial species (Faecalibacterium prausnitzii, Clostridioides difficile, Akkermansia muciniphila, Bacteroides vulgatus, Streptococcus thermophilus, and Shiga toxin-producing Escherichia coli (stx1 gene positive)) were chosen as bacterial biomarkers for further quantification via qPCR in all samples. Primers were designed using the IDT SciTools® platform (Owczarzy et al., 2008). Primer sequences for each species and total bacterial quantification are shown in Table 1. In each run both positive and negative controls were added to qPCR-plates. For positive control A. muciniphila (ATCC BAA-835), B. vulgatus (ATCC 8482), S. thermophilus (ATCC 19258), F. prausnitzii (ATCC 27766), E. coli (ATCC 43890), and C. difficile (ATCC 9689) were utilized. A total of 2.5 µl of distilled water was used as a negative control in each run.

The reactions were conducted with 2.5 µl DNA, 1.9 µl PCR-grade water, 0.3 µl primer, and 5 µl LightCycler® 480 SYBR Green I Master enzyme (Roche Applied Science, Penzburg, Germany) in a total volume of 10 µl. All reactions were carried out with triple replicates on a LightCycler® 480 II (Roche Applied Science, Penzburg, Germany) qPCR machine. Each sample and species is assayed at least twice. The qPCR reaction was 95 °C for 10 min with initial denaturation followed by 50 cycles of 95 °C for 10 s, 58 °C for 15 s, and 72 °C for 15 s. Melting curve analysis for the qPCR products was performed under the following conditions: 95 °C for 5 s, 63 °C for 1 min and a denaturing temperature ramp from 63 to 97 °C with a rate of 0.11 °C/s. Amplification and melting curves for each sample were obtained using Absolute Quantitation/Second Derivative and Tm Calling analysis modes in the LightCycler® 480 II Software v.1.5.

Table 1 Primers used for detection and quantification of selected bacterial species.

Target Bacteria	Genome region	DNA sequences of the primers (5′–3′)	Product size (bp)	References	
F. prausnitzii	16S rRNA	Forward	GGAGGAAGAAGGTCTTCGG	248	Fujimoto et al. (2013)	
Reverse	AATTCCGCCTACCTCTGCACT	
C. difficile	16S rRNA	Forward	TTGAGCGATTTACTTCGGTAAAGA	157	Balamurugan, Balaji & Ramakrishna (2008)	
Reverse	CCATCCTGTACTGGCTCACCT	
A. muciniphila	Genomic location	Forward	GAAGACGGAGGACGGAACT	126	Osman et al. (2021)	
Reverse	GCGGATTGCTGACGAAGG	
B. vulgatus	16S rRNA	Forward	GCATCATGAGTCCGCATGTTC	287	Ishaq et al. (2021)	
Reverse	TCCATACCCGACTTTATTCCTT	
S. thermophilus	groL	Forward	GCTGTGGAAGAGCTTAAAGTC	138	Falentin et al. (2012)	
Reverse	ACCATCATTACCAACGCGT	
Shiga toxin-producingE. coli	stx1	Forward	GTCACAGTAACAAACCGTAACA	95	Fernandez et al. (2020)	
Reverse	TCGTTGACTACTTCTTATCTGGA	
Total bacteria	16S rRNA	Forward	TCCTACGGGAGGCAGCAGT	466	Balamurugan, Balaji & Ramakrishna (2008)	
Reverse	GGACTACCAGGGTATCTAATCCTGTT	

SYBR Green dye fluorescence intensity was used for quantification. The target bacterial DNA concentration correlated with the threshold cycle number (Ct), the cycle number at which fluorescence signal was first detected. Roche LightCycler® 480 System melting curve program analysis is used for the confirmation of success of qPCR reactions (Simenc & Potocnik, 2011). Data analysis was conducted using both the 2−ΔCt, and 2−ΔΔCt methods (Livak & Schmittgen, 2001). Target microorganisms were considered as a target while total bacteria measurement was used as a reference (Navidshad, Liang & Jahromi, 2012). Relative abundance of bacteria was expressed as log2 transformed fold change values, and calculated according to the following formulas.

Relative abundance of target bacteria species with respect to abundance of total bacteria: 2−ΔCt=2- (Ct of target bacteria−Ct of total bacteria).

Fold change of relative abundance of target bacteria in IBD patients compared to healthy controls 2−ΔΔCt=2- [(Ct of target bacteria−Ct of total bacteria) patient - (Ct of target bacteria−Ct of total bacteria) control].

To verify PCR efficiency, standard curves were generated by 10-fold dilutions of bacterial DNA for all primer sets. In all sets, qPCR efficiency was >90% and calculated by E= 10(−1/slope)−1 equation. According to the serial dilutions, the limit of detection of qPCR assays was 10–100 copies.

Statistical analysis

Distribution of candidate gene’s genotypes among UC, CD, and control groups was compared with Chi-square categorical analyses. Shapiro–Wilk test was used to test the normality assumption of numeric variables. All microbial diversity and quantity parameter estimates violated normality assumption, so groups were compared using the non-parametric Kruskal–Wallis and pairwise Wilcoxon rank-sum tests. P-value less than 0.05 was assessed as statistically significant. All statistical analyzes were performed with R software (Version 4.0.5) (https://www.r-project.org). We also estimated the statistical power to detect a 10% change with 95% confidence in the numeric abundances of A. muciniphila, B. vulgatus, S. thermophilus, and F. prausnitzii in UC and CD groups based on qPCR relative abundance estimates. Statistical power calculations followed formulations in Cohen (1988) as implemented in the pwr package (https://cran.r-project.org/web/packages/pwr/index.html) of R software. Experimentally determined relative abundance means and standard errors of A. muciniphila, B. vulgatus, S. thermophilus, and F. prausnitzii are used for statistical power calculations, where Type I error probability (alpha) is set to 0.05.

Results

Patient characteristics

UC and CD patients consisted of similar age and sex groups, however, included a variety of disease locations and disease activity phenotypes (Table 2, Table S2). Distribution of genotypic frequencies of ATG16L1, IL23R and NOD2 variants in UC and CD patients was similar (Table S3), suggesting similar IBD genetic risk profile in these patient groups.

Identification of fecal microbiota profile in the IBD patients and controls

An average of 148,545 reads per sample (range 125,105–172,435) were generated for the nine fecal samples (three samples from each of CD, UC, and control groups). Eight phyla, 15 classes, 23 orders, 43 families, 96 genus, and 233 species were represented in all sequences based on a 97% similarity level. The most abundant phyla in CD samples were Firmicutes (36.22%), Proteobacteria (29.22%), Verrucomicrobia (25.79%), Bacteroidetes (8.68%), Actinobacteria (0.07%), Fusobacteria (0.01%), and Synergistetes (0.01%). Similarly in UC samples, Proteobacteria (45.63%), Firmicutes (29.21%), Bacteroidetes (10.14%), Fusobacteria (9.98), Actinobacteria (2.80%), and Verrucomicrobia (2.25%) were observed (Fig. 1A). The most common phyla in the control samples were Bacteroidetes (62.63%), followed by Firmicutes (31.47%), Proteobacteria (4.01%), Actinobacteria (1.81%), Fusobacteria (0.04%), Lentisphaerae (0.04%), and Verrucomicrobia (0.01%) (Fig. 1A). Despite small sample size Kruskal–Wallis tests showed abundance difference for Proteobacteria (p = 0.03), and Firmicutes (p = 0.07) between IBD and control samples. At the family level, significant abundance differences for Bacteroidaceae (p = 0.006), Rikenellaceae (p = 0.03), Acidaminococcaceae (p = 0.05), Victivallaceae (p = 0.03), and Enterobacteriaceae (p = 0.03) were observed between IBD and control samples (Fig. 1B).

Table 2 Control, patient group, and disease characteristics.

	Crohn’s disease (%) N = 12	Ulcerative colitis (%) N = 6	Control (%) N = 4	
Median Age, years (25%, 75%)	51 (48, 56)	47 (41, 50)	24 (22, 24)	
Sex (%)				
Female	6 (50)	4 (67)	3 (75)	
Male	6 (50)	2 (33)	1 (25)	
Median disease duration, years (25%, 75%)	7 (2, 9)	6 (1, 14)	–	
Smoking History				
Yes (%)	2 (17)	2 (33)	1 (25)	
Disease localization (%)				
Ileal	1 (8)	–	–	
Colonic	4 (33)	–	–	
Ileocolonic	3 (25)	–	–	
Surgery	3 (25)	–	–	
Penetrating perianal disease	3 (25)	–	–	
Distal colitis	–	1 (17)	–	
Left colitis	–	2 (33)	–	
Pancolitis	–	3 (50)	–	
Treatment (%)			–	
Biologics	9 (75)	4 (67)	–	
Non-biologics	3 (25)	2 (33)	–	
Median Mayo Score (25%, 75%)	–	5 (4, 6)	–	
Harvey-Bradshaw Index (25%, 75%)	6.5 (3, 7.5)	–	–	
C-reactive protein (CRP mg/dL)				
<5 (%)	6 (50)	4 (67)	–	
>5 (%)	6 (50)	2 (33)	–	
Notes.

CRP level at fecal sample collection time indicating severity of disease activity.

Figure 1 Microbiota composition comparisons.

Phylum level (A), and family level (B) taxon relative abundance comparisons among Crohn’s disease (CD), Ulcerative colitis (UC), and control groups. (C) Venn diagram illustrating the number of common shared and unique bacterial species observed in Crohn’s disease, Ulcerative colitis, and control groups.

Reduction in alpha diversity estimates in the IBD samples (UC and CD groups) compared to control samples was observed (Fig. S1), however only the Shannon diversity estimate was significantly lower in the CD group compared to the control group (p = 0.04). Overall microbial dysbiosis commonly reported in IBD, and greater dysbiosis in CD is captured in the study.

Identification and quantification of biomarker bacterial species

Fecal microbiota analyses identified not only the taxa unique for UC, CD, and control groups, but also common bacterial species among the groups (Table S4). In fact, 29 bacterial species that were found in all three groups cumulatively constituted nearly 60% of all taxon assigned reads identified in the fecal microbiota (Fig. 1C). Among the 29 bacterial species found in all three groups, Faecalibacterium prausnitzii, Akkermansia muciniphila, Bacteroides vulgatus, Streptococcus thermophilus, and Escherichia coli were the most common ones making up 84% of the reads assigned to these 29 bacterial species.

Shiga toxin-producing Escherichia coli is a proinflammatory bacteria, reported to show increasing abundance in IBD patients, whereas harmful or beneficial association of Bacteroides vulgatus in IBD is less certain (da Silva Santos et al., 2015; Palmela et al., 2018; Zafar & Saier Jr, 2021). Faecalibacterium prausnitzii, Streptococcus thermophilus, and Akkermansia muciniphila are beneficial bacteria, reported to be reduced in IBD patients (Sokol et al., 2008; Prosberg et al., 2016; Zafar & Saier Jr, 2021). These five species are chosen as potential biomarker bacteria for further quantification in all UC and CD samples. Because higher abundance of pathogenic Clostridioides difficile is also reported to be associated with IBD (Issa et al., 2007), C. difficile is also chosen to be further quantified in the IBD samples.

Quantitative real-time PCR analyses were conducted with primers specific to each target species (Table 1). Specificity of primers and success of the qPCR reactions were checked by melting curve analyses (Fig. S2). After quantification of six selected species in all twenty two samples by qPCR, the relative abundance of candidate species compared to total bacteria amount is calculated by the 2−ΔCt method for each sample. Reduction in the relative abundance of beneficial species F. prausnitzii and A. muciniphila, in UC and CD samples compared to total bacteria amount was evident (Fig. 2). The relative abundance of B. vulgatus was higher in the control samples (Fig. 2). Interestingly, the relative abundance of S. thermophilus was highest in the UC samples (Fig. 2). Shiga toxin-producing E. coli and C. difficile was observed in only three of the CD patients. The next analyses compared the fold change of A. muciniphila, B. vulgatus, S. thermophilus, and F. prausnitzii in IBD samples with respect to control samples calculated by the 2−ΔΔCt method. Significant reduction in A. muciniphila, S. thermophilus, and F. prausnitzii in combined IBD samples compared to controls was observed (Fig. 3). However, fold change values of these four species were similar in the UC and CD samples (Fig. 3).

Figure 2 Comparison of relative abundance of selected common bacteria.

Relative abundance of (A) Akkermansia muciniphila, (B) Bacteroides vulgatus, (C) Streptococcus thermophiles, (D) Faecalibacterium prausnitzii with respect to total bacteria quantified by qPCR within each group. Y-axis values are the ratio of target bacteria abundance to total bacteria abundance. UC, Ulcerative colitis; CD, Crohn’s disease.

Figure 3 Comparison of relative abundance change of Akkermansia muciniphila, Bacteroides vulgatus, Streptococcus thermophiles, and Faecalibacterium prausnitzii.

Comparison of log2 fold change of relative abundance of Akkermansia muciniphila, Bacteroides vulgatus, Streptococcus thermophiles, and Faecalibacterium prausnitzii with respect to control samples in (A) IBD (UC and CD groups together), (B) in ulcerative colitis (UC) and Crohn’s disease (CD) samples. P values calculated from non-parametric Wilcoxon rank-sum tests.

Effect of disease course and treatment on common bacterial species is also examined. Patients on biologics treatment presented with higher reduction in the relative abundance of beneficial F. prausnitzii compared to patients who are not on biologics (mean log reduction 5.8 vs. 2.1, p = 0.04). Within the CD group, patients who had surgery showed higher reduction in the relative abundance of A. muciniphila compared to patients who did not have surgery (mean log reduction 21 vs. 15, p = 0.02). Moreover, patients with penetrating perianal CD (more severe version of CD) again showed higher reduction in the relative abundance of A. muciniphila (mean log reduction 21 vs. 16, p = 0.05), and increase in the relative abundance of B. vulgatus (mean log increase 10 vs. 5, p = 0.05). Within the UC group, patients with CRP levels higher than five showed higher reduction in the relative abundance of B. vulgatus compared to patients with CRP levels less than five (mean log reduction 14 vs. 4, p = 0.05).

Utility of NGS (Next Generation Sequencing) and qPCR results

None of the correlations between the NGS read and the qPCR results of either individual selected bacteria or total bacteria was statistically significant. Highest correlation was observed for total bacteria results (Adjusted R2 = 0.14) but it was not significant (p = 0.18). Lack of congruency between the two techniques was in part due to great variation in NGS read results. The standard deviation of NGS reads ranged from a lowest of 724 for A. muciniphila to 4,352 for S. thermophilus. Contrary, the standard deviation of qPCR results ranged from a lowest of 6.3 for A. muciniphila to 14.4 for B. vulgatus, showing orders of magnitude smaller variance in the qPCR results compared to NGS results. However, the lower variation in qPCR assays is also due to the concomitant characteristics of the qPCR technique and range of the data generated. Moreover, we did not incorporate unique molecular identifier (UMI) analysis in NGS data that enables identification of PCR duplicates into sequencing libraries, which can introduce noise in the read counts.

The much lower variation inherent in qPCR results suggests higher accuracy, precision, and higher clinical utility for the qPCR technique compared to NGS based amplicon sequencing approach especially with smaller sample sizes. Statistical power calculations based on mean and standard deviation estimates from our experimental data showed that qPCR results have over 80% statistical power to detect a minimum 10% difference in candidate species abundance with less than 200 samples (Fig. 4). Such high statistical power is nearly impossible to achieve with NGS methods. Given that IBD clinics have hundreds of patients, and microbiota alterations as a response to therapy/interventions are usually much higher than 10%, qPCR surveillance of candidate bacteria has good clinical potential to monitor microbiota response through disease and therapy course.

Figure 4 Relationship between sample size and statistical power.

Relationship between sample size and statistical power to detect a 10% change with 95% confidence in relative abundance of (A) Akkermansia muciniphila, (B) Bacteroides vulgatus, (C) Streptococcus thermophiles, (D) Faecalibacterium prausnitzii in ulcerative colitis (UC) and Crohn disease (CD) samples. Power calculations are based on the mean and standard deviation estimates from experimental results for each bacterial species. For B. vulgatus power to detect 20% change numbers are presented.

Discussion

We aimed to test whether qPCR analysis of candidate common bacterial species identified from a patient’s individual fecal microbiome can be used as a fast and reliable personalized microbial biomarker for efficient monitoring of IBD. NGS based fecal microbiota analyses followed by targeted qPCR analyses of candidate common bacterial species showed to be an efficient and reliable method for monitoring of disease status in IBD patients.

Firstly, thorough microbiota analyses identified bacterial taxa in UC, CD, and controls at the species level resolution. Microbiota profiles obtained in this study was similar to the reported profiles in the literature agreeing with dysbiosis in CD and UC patients (Kostic, Xavier & Gevers, 2014; Pascal et al., 2017). Based on the findings of microbiota analyses, bacterial species reported to be positively and negatively associated with IBD are chosen as candidate biomarker species. However, rather than targeting rare species that are observed in UC or CD, we primarily focused on high abundance bacterial species that are commonly observed not only in IBD but also in healthy controls. Faecalibacterium prausnitzii, Akkermansia muciniphila, and Streptococcus thermophilus are gut bacteria with anti-inflammatory properties suggested to be important in gut homeostasis. Their abundance is reported to be reduced in IBD patients (Prosberg et al., 2016; Pascal et al., 2017; Zafar & Saier Jr, 2021). Our qPCR analyses also showed significant reduction in abundance of these beneficial bacteria in the IBD samples compared to healthy control samples. Moreover, the reduction in the relative abundance of these bacteria was greater in patients with worse disease progression such as in patients with penetrating CD, higher CRP levels (higher inflammation), and patients who require biologics treatment. Clostridium difficile, and Shiga toxin-producing Escherichia coli, on the other hand, are harmful bacteria with mucus degrading, invasive, pro-inflammatory properties reported to be in high abundance in IBD patients (Issa et al., 2007; da Silva Santos et al., 2015; Prosberg et al., 2016; Palmela et al., 2018). In our qPCR analyses C. difficile and Shiga toxin-producing E. coli were only observed in CD samples. These results show that qPCR results are specific to targeted species, are in agreement with literature reports, and therefore are reliable.

IBD microbiota studies are advancing from just reporting descriptive microbiota changes to examining correlations between microbiota profiles, and IBD disease activity, course, and treatment response. Recently, certain bacterial taxa (such as Clostridiales, Eubacteria, Bifidobacteria) are suggested to be associated with treatment response, relapse, and disease progression (Rajca et al., 2014; Kolho et al., 2015; Zhou et al., 2018). However, in these studies, the taxonomic resolution is course and not at the species level. With appropriate species specific primers, qPCR analysis can be highly sensitive and accurate identifying the altered species in IBD. Species level information can be used for better association tests and predictions with respect to clinical and treatment phenotypes. In addition, statistical modelling and (clinical) interpretation of multivariate microbiota (microbiome) data with respect to a (clinical) phenotype is much harder compared to univariate species specific statistical association analysis, limiting the clinical usefulness of multivariate microbiota data.

Although NGS approach is proposed to identify rare taxa that can be unique to UC or CD, the clinical utility of microbiota data generated by NGS based 16S rRNA gene amplicon sequencing is still debated. Firstly, NGS based methods are more expensive, time consuming, and require bioinformatics infrastructure and expertise. In addition, methodological issues (due to PCR artifacts, sequencing platform, DNA isolation and contamination, etc.), huge variation in the number of sequence reads, different microbiota results generated even analyzing the same sample (Hiergeist et al., 2016; Boers, Jansen & Hays, 2019) hinder usage of NGS based microbiota results in monitoring disease status and course in IBD patients. In this study, the variance associated with 16S rRNA gene NGS reads was also much higher compared to the relative abundance variances estimated from qPCR data, making 16S rRNA gene NGS data more noisy for statistical comparisons.

Some bacterial species and strains have multiple 16S rRNA gene copies in their genomes making the16S rRNA gene NGS based estimates of relative abundance and representation of these taxa in the microbiome erroneous (Vetrovsky & Baldrian, 2013; Louca, Doebeli & Parfrey, 2018). Because all taxon identification databases are based on 16S rRNA gene sequence, NGS based methods do not have the alternative of targeting other genome regions. A possible solution is adapting a metagenomics approach, and sequencing whole genomes. However, metagenomics is even harder, more problematic, time consuming, and much more expensive than 16S rRNA gene amplicon sequencing. On the other hand, in a qPCR approach one can easily target genome regions other than the 16S rRNA gene, and alleviate possible distorted relative abundance estimates due to multiple 16S rRNA gene copies.

There are several limitations of the study. The sample size is small, and longitudinal sampling of microbiota is not available. Although the sample size is small, patients with diverse disease location and activity phenotype characteristics are involved in the study. So, the results of the qPCR approach are not just specific to a subgroups of UC or CD patients, but can be generalizable to broader IBD patients. There are several other pathological E. coli associated with IBD, however we only focused on Shiga toxin-producing E. coli, quantification of E. coli in general could be more informative. Moreover, IBD has a genetic component, and genetic variants in NOD2, ATG16L1, and IL23R are reported to be associated with highest IBD risk (McGovern, Kugathasan & Cho, 2015). We tested whether genetics can be a confounder of the results, but IBD genetic risk profiles of UC and CD groups were similar. There was no single dominant IBD risk genotype in the UC and CD groups. So a genetic stratification confounding the results is unlikely. We acknowledge that there can be other genetic factors that can influence the course of IBD and treatment response. These additional genetic factors can be considered in the future studies, if deemed necessary by the medical community.

Conclusions

In conclusion, qPCR analysis of common candidate bacterial species identified from a patient’s individual fecal microbiome can be used as a fast and reliable personalized microbial biomarker for efficient monitoring of IBD. Moreover, the relative abundance of these common bacterial species showed association with worse disease progression in IBD.

Our results should stimulate further studies adopting personalized microbiota based qPCR analysis of targeted bacterial species longitudinally sampled from larger sized cohorts of IBD patients.

Supplemental Information

Supplemental Information 1 Shannon, Simpson, Ace, and Chao1 alpha diversity estimate comparisons among Crohn (CD), Ulcerative colitis (UC), and control groups

Click here for additional data file.

Supplemental Information 2 Melting curve analyses of (A) Streptococcus thermophiles, Bacteroides vulgatus; (B) Pathogenic Escherichia coli, Akkermansia muciniphila; (C) Faecalibacterium prausnitzii, Clostridium difficile, and total bacteria

Click here for additional data file.

Supplemental Information 3 Primers targeting IBD related genes and their variants

Click here for additional data file.

Supplemental Information 4 Control, patient group, and disease characteristics for samples included in 16S rRNA gene NGS analysis

Click here for additional data file.

Supplemental Information 5 Distribution of genotypic frequencies of ATG16L1, IL23R and NOD2 variants among CD and UC patients

Click here for additional data file.

Supplemental Information 6 Bacterial species unique to and shared between Ulcerative Colitis (UC), Crohn disease (CD), and Control groups

Click here for additional data file.

Supplemental Information 7 Raw patient data used for all analyses

Click here for additional data file.

Supplemental Information 8 Raw Species Level Taxon Data

Raw species level taxon table used for NGS based microbiota analyses

Click here for additional data file.

Supplemental Information 9 Reverse primer reads

Microbiome next generation sequence data of 16S rRNA reads.

Click here for additional data file.

Supplemental Information 10 Forward primer reads

Microbiome next generation sequence data of 16S rRNA reads.

Click here for additional data file.

Additional Information and Declarations

Competing Interests

Author Contributions

Human Ethics

Data Availability

Efe Sezgin is an Academic Editor for PeerJ.

Efe Sezgin conceived and designed the experiments, performed the experiments, analyzed the data, prepared figures and/or tables, authored or reviewed drafts of the article, and approved the final draft.

Gamze Terlemez conceived and designed the experiments, performed the experiments, analyzed the data, prepared figures and/or tables, authored or reviewed drafts of the article, and approved the final draft.

Berkay Bozkurt performed the experiments, analyzed the data, prepared figures and/or tables, and approved the final draft.

Göksel Bengi conceived and designed the experiments, authored or reviewed drafts of the article, and approved the final draft.

Hale Akpinar conceived and designed the experiments, authored or reviewed drafts of the article, and approved the final draft.

İlker Büyüktorun performed the experiments, prepared figures and/or tables, and approved the final draft.

The following information was supplied relating to ethical approvals (i.e., approving body and any reference numbers):

Dokuz Eylul University.

The following information was supplied regarding data availability:

BioProject ID: PRJNA853045.

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
