# Peer review of "Quantitative real-time PCR analysis of bacterial biomarkers enable fast and accurate monitoring in inflammatory bowel disease"

_PeerJ, doi:10.7717/peerj.14217_

## Round 0.1 · original submission · Major Revisions

· Academic Editor

Major Revisions

Manuscript ID 2022:06:74801:0:2 entitled "Quantitative real-time PCR analysis of bacterial biomarkers enable fast and accurate monitoring in inflammatory bowel disease" which you submitted to PeerJ has been reviewed. The comments of the referee(s) are included at the bottom of this letter.

Please note that submitting a revision of your manuscript does not guarantee eventual acceptance, and that your revision may be subject to re-review by the referee(s) before a decision is rendered. Please submit your revised manuscript with the changes you have made in response to the reviewer's suggestions.

Reviewer 1 ·

Basic reporting

The article by Terlemez et al. identifies microbial biomarkers in faecal samples of IBD from Turkey by means of NGS and further quantifies them by qPCR. The selected biomarkers are chosen by being species found in all groups of subjects analysed.

The manuscript is written in clear English and can be read easily. Minor grammatical errors have been found. The structure of the article includes the standard sections according to the Instructions for Authors.

From my point of view, the background is quite narrow and it is not fully demonstrated how the work fits into the broader field of knowledge as expected by PeerJ. Specifically, the authors aim to use species common in all subjects regardless of their gut health/disease status, but the applicability is focused in IBD participants from Turkey and not validated with cohorts apart from that of the study.
In addition, the authors claim as novelty the quantification of bacterial biomarkers by qPCR to aid in IBD diagnostics. However, this has been extensively explored previously, including NGS and quantification by qPCR of some of the species analyzed by the authors. Examples of some key references that are missing are:
- Pascal V, et al. Gut 2017;66:813–822. DOI: 10.1136/gutjnl-2016-313235
- López-Siles M, et al. Front Cell Infect Microbiol. 2018 Sep 7;8:281. doi: 10.3389/fcimb.2018.00281.
Therefore, these claims should be softened and adjusted to the novelty of the manuscript.

Figures are relevant to the content of the article, appropriately described and labelled. However, resolution should be improved for Figures 1, 2 and 3. In addition, for Figure 1, names of Families are too small to be seen clearly. Similarly, labels for Y and X axis in Figure 2 are difficult to see clearly. To increase the size of Figure 3 including labels above each box is also advised. In addition, I suggest to delete the line joining means between groups of patients to make plots more clean. A combination of dot and box plots can be drawn instead.

I confirm that raw data has been submitted by the authors and I could access to it.

The submission represents a coherent body of work and has not been inappropriately subdivided. However, the main points to address are that (i) results presented are not enough to proof the hypothesis formulated by the authors and (ii) the main claim does not represent a major novelty upon current knowledge in the field. First, because it is claimed that qPCR of bacterial biomarkers can accurately monitor IBD but previous studies already demonstrated similar results. Secondly the findings of the authors have been tested in a limited number of subjects, highly diverse in characteristics and no corroboration of their findings with an independent cohort has been carried out. It is not clear if the discovery cohort (NGS) is included among the patients further used for qPCR, please clarify. In any case, the number of subjects engaged is very limited. Therefore, I suggest that the statement of applicability as biomarkers to assist in IBD should be softened throughout the manuscript, upon validation among different cohorts is performed. Lastly, they state that “Unique to this study, rather than focusing on rare taxa distinct to UC or CD groups, common high abundance bacterial species observed in all three groups, and reported to be associated with IBD are chosen as candidate marker species.” Previous studies have already focused on the use of these species as biomarkers (see suggested references above) and some of these species are not high abundance bacterial species observed in all three groups (for instance IBD patients feature a depletion in F. prausnitzii). Raw data of the NGS provided by the authors also demonstrates that (F. prausnitzii was detected only in 1 UC and 1 CD, similarly, Akkermansia was detected in 1 CD,2 UC and 1 control). I strongly recommend that the article is rewritten considering previous knowledge and reformulating the novel knowledge provided (mainly the inclusion of S. thermophilus as biomarker and applicability of previous biomarkers in Turkey population).

Experimental design

It is an original primary research with a question well defined, relevant and meaningful. However, some advances in the knowledge gap identified by the authors have been already studied by the scientific community, and they are not mentioned in the introduction. Specifically, they propose qPCR as a faster, cheaper, more accurate techniques available equipment in hospitals for IBD diagnostics and to test microbial biomarkers for efficient monitoring of IBD. In this sense, over the past decade several works indicating usefulness microbial biomarkers by means of qPCR are available in the literature. For example:
• Machiels K, et al. Gut 2013; 63 1275-1283. doi: 10.1136/gutjnl-2013-304833.
• López Siles, M.; et al. International Journal of Medical Microbiology 2014. 304: 464-475. doi: 10.1016/j.ijmm.2014.02.009
• Zhou et al. Medicine 2016 Sep;95(39):e5019. doi: 10.1097/MD.0000000000005019.
• Mondot et al Gut 2016 Jun;65(6):954-62. doi: 10.1136/gutjnl-2015-309184.
• López Siles, M.; et al. World Journal of Gastrointestinal Pathophysiology 2020 May 12;11(3):64-77. doi: 10.4291/wjgp.v11.i3.64.
Therefore, I suggest the authors to (i) complete the introduction by including previous knowledge of qPCR assays to monitor IBD and (ii) adapt accordingly the knowledge gap to fill (i.e. to assess if those previously tested are applicable in Turkey cohorts and additional information provided by S. thermophilus, whose usefulness as biomarker has been less explored).

Information concerning ethical committee approval of the study is included. In general, the investigation concurs with technical standards and methods are well documented except for qPCR assays:
• Please indicate which strain of each microorganism were used as positive controls in each reaction, and also if negative controls were performed in each run.
• Also limit of detection, and efficiency are not provided. I suggest to consider the MIQE – Minimum Information for Publication of Quantitative Real-Time PCR Experiments guideline for reference (Bustin A.S. et al. Clinical Chemistry, 2009 55 (4): 611–622, https://doi.org/10.1373/clinchem.2008.112797) and complete missing information.
• According to Supplemental Fiugre 2, it seems that the microorganisms were quantified at once in a multiplex qPCR. It was not clear to me in the main text, please specify.

Validity of the findings

Given the coincidence of microbial biomarkers analyzed in this study and previous ones, this article falls more in the category of a replication study but in a cohort from Turkey. This is of interest as few studies have been carried out in this population previously. However, the focus of the manuscript should be re-written in this sense.

In addition, some points in the results and the derived conclusions need to be addressed:
- L215-216 and Figure 4. Please, develop further how these calculations were carried out. To assess the utility of qPCR data for diagnostics, ROC curve analysis should have been carried out and specificity, sensitivity and accuracy calculated.
- L256-The authors selected to quantify shiga toxin-producing E. coli. However, other pathotypes such as adherent-invasive E. coli have been associated to Croh’s disease. Because the identification by NGS does not discriminate shiga toxin producing E. coli, and other E.coli pathotypes have been related to IBD, it would be more accurate to quantify E. coli in general.
- L284- I think in Table 2 the number of subjects with penetrating perianal CD is missing. Please provide this information to see the number of patients that support this finding.
- L388. The authors indicate that the relative abundance of the species show association with the severity of IBD. However, it is not clear what they include in “severity”. Does it refer to CRP? Active/inactive disease? Penetrating phenotype? An additional figure or table comparing abundance of the six bacterial indicators by each clinical variable would be of help to illustrate this point.
- The authors conclude that their results will prompt modelling disease course, and predicting therapy response paving the way for better personalized treatment options for IBD patients. Because analysis of different treatments have not been carried out and measuring therapy response, this conclusion cannot be drawn from their data and rather should be proposed in the discussion as future studies.

Additional comments

The following minor points were identified:
-Table 1: Genome region should be 16S rRNA gene instead of 16S Ribosomal
-Table 2: Smoking history, treatment and CRP do not have units. Do numbers represent % (n)?
- L33- Should be 16SrRNA gene
-L180, L263- Clostridium difficile has been re-classified as Clostridioides difficile
-L 245- The “evident reduction in alpha diversity” would be better supported by including if differences are significant or not, and adding a p-value
-L300- The lower variation in qPCR assays is also due to the concomitant characteristics of the technique (the linear rang span is lower than NGS reads). This has not been considered. Please coment further on this characteristic.
-L260: 16S NGS should be 16S rRNA gene NGS
- Table 2. Include data from control cohort and also differentiate data from NGS cohort and that for qPCR analysis.
- L346- It is not clear yet whether altered microbiota is a cause or a consequence of IBD, therefore qPCR analysis would not identify the causal species, rather than alterations. Please rephrase.
- L361- The authors point that different copy numbers of 16S rRNA gene among species is a problem for NGS analysis of microbiota. However, the majority of the qPCR assays used in this study also target that gene. Please comment on the reason why it is not a problem in qPCR assays but it is for NGS.

Reviewer 2 ·

Basic reporting

This article is well written with clear and unambiguous professional English. All materials are clearly laid out. Figures are relevant, well labeled and described. I suggest that the authors also provide a table listing bacteria species common and unique to Ulcerative colitis, Crohn's disease and healthy individuals.

Experimental design

This article used next generation sequencing of the 16S rRNA region to investigate the bacterial profile from a subset of UC, CD, and control samples. While most methods in this manuscript are described with sufficient details and information to replicate, it is unclear how sequencing depth was determined and whether it is the appropriate and sufficient depth for this type of analysis. Related studies should be referenced for your justification in this part.

Validity of the findings

The goal of this study is to identify microbial biomarkers for efficient monitoring of disease course in Inflammatory bowel diseases. The authors used Next Generation Sequencing(NGS) to first characterize the bacterial community and then focused on differences in the composition of common bacterial species. One of the key points reported and discussed in this study is the comparison between these two methods of bacterial profiling, NGS and qPCR method. The authors observed no correlation between NGS read and the qPCR results due to very noisy data from NGS compared to qPCR, which is not surprising due to their experimental design. Next Generation Sequencing in this study did not incorporate UMIs, enabling the accurate bioinformatic identification of PCR duplicates, into sequencing libraries. Therefore, the use of NGS here is not for quantitative analysis. I suggest that the authors revisit part 3.4 because the current comparison is not very compelling. Moreover, NGS is supposed to provide higher discovery power to detect novel and rare species of bacteria associated with inflammatory bowel disease, but I found the authors rarely discussed their findings of rare taxa distinct to UC or CD groups. The rare taxa can potentially be additional biomarkers to be considered in conjunction with the relative abundance of common bacterial species to monitor inflammatory bowel disease.

Reviewer 3 ·

Basic reporting

Dear Editor, dear Authors,
as requested, I reviewed the research article entitled “Quantitative real-time PCR analysis of bacterial biomarkers enable fast and accurate monitoring in inflammatory bowel disease (#74801)” by Gamze Terlemez, Berkay Bozkurt, Göksel Bengi, Hale Akpnar, lker Büyüktorun, and Efe Sezgin. The article deals with the identification of microbial biomarkers in fecal microbiome from a cohort of Turkish IBD patients. Authors describe evidence of lowering of beneficial species such as Faecalibacterium prausnitzii, Akkermansia muciniphila, and Streptococcus thermophilus in IBD compared to healthy control samples, also indicating correlation with IBD severity.
The paper is well professionally written, clear and unambiguous. The Introduction section suggests scientific mastery of the topic by the authors, being at the same time concise and complete in the information. The overall article structure is well and rationally conceived; the overall results are coherent and clearly exposed by data, graphs and figures.

Experimental design

The experimental design diligently follows the authors' goal, and the results represent a very useful contribution to their ever-evolving field of research. Methods are as expected and are reported with sufficient accuracy for repetition.

Validity of the findings

The results represent a very useful contribution within their ever-evolving field of research. Conclusions are sufficiently well stated according to results.

Additional comments

N/A

---

## Round 0.2 · Minor Revisions

· Academic Editor

Minor Revisions

It is my opinion as the Academic Editor for your article - Quantitative real-time PCR analysis of bacterial biomarkers enable fast and accurate monitoring in inflammatory bowel disease - that it requires a number of Minor Revisions as per the comments of Reviewer 1

Reviewer 1 ·

Basic reporting

The manuscript has improved substantially and it is very much appreciated the effort that the authors have made to solve the points risen.

However, some issues have not fully solved, as listed below:
- While the authors have softened the claims in the manuscript, these changes are not reflected in the abstract. Some examples from abstract vs main text are:

Example #1:
L34 (abstract): Unique to this study, rather than focusing on rare taxa distinct to UC or CD groups, common high abundance bacterial species observed in all three groups, and reported to be associated with IBD are chosen as candidate marker species.
L350 (main text): However, rather than targeting rare unique species that are specific to just UC or CD, we primarily focused on high abundance bacterial species that are commonly observed not only in IBD but also in healthy controls.

Example #2:
L42 (abstract): Personalized microbiota based qPCR analysis of target common bacterial species can be a powerful, cost and time efficient approach for monitoring disease status and identify better personalized treatment options for IBD patients.
L421 (main text): In future studies, such quantitative relative abundance data can be amenable to modelling disease course, and predicting therapy response paving the way for better personalized treatment options for IBD patients.

Please modify the abstract accordingly and rewrite focusing in the novelty of the manuscript.

-While Figures in the main text have been considerably improved, those changes have not been applied to supplemental Figures. For example, the same modifications carried out for Figure 3 could be used to improve Supplementary Figure 1. Please, modify accordingly.

Experimental design

No comment

Validity of the findings

No comment

Additional comments

Other minor points detected while revising are:
-There are still some “16SrRNA” that should be “16SrRNA gene” (Ex. L166, L379). Please revise throughout the manuscript.
-L232, L236:”S, thermophiles” I think should be “S. thermophilus” (dot instead of coma after genus abreviation, and –us ending). Please revise the full text.

---

## Round 0.3 · Minor Revisions

· Academic Editor

Minor Revisions

Dear Dr. Sezgin,

Thank you for your submission to PeerJ. As per Reviewer 1, I suggest you attend to this final minor revision.

Reviewer 1 ·

Basic reporting

No comment

Experimental design

No comment

Validity of the findings

No comment

Additional comments

It is appreciated that the authors have corrected the additional points raised. I am afraid that I have to indicate an additional issue for their consideration.
Specifically in the last version submitted they added in the abstract the statement "And, S. thermophilus may be useful biomarker in IBD." but this is not supported by findings in Figure 3. I would delete this sentence. If they want to highlight another novel point of they work, it could be the validation of the previously identified biomarkers in Turkey population.

---

## Round 0.4 · accepted · Accept

· Academic Editor

Accept

I am writing to inform you that your manuscript - Quantitative real-time PCR analysis of bacterial biomarkers enable fast and accurate monitoring in inflammatory bowel disease - has been Accepted for publication.